# Efficient Convolutional Neural Network Model for the Taxonomy and Sex Identification of Three Phlebotomine Sandfly Species (Diptera, Psychodidae, and Phlebotominae)

**DOI:** 10.3390/ani14243712

**Published:** 2024-12-23

**Authors:** Mohammad Fraiwan

**Affiliations:** Department of Computer Engineering, Jordan University of Science and Technology, P.O. Box 3030, Irbid 22110, Jordan; mafraiwan@just.edu.jo

**Keywords:** taxonomy, sex identification, convolutional neural networks, *Phlebotomus alexandri*, *Phlebotomus sergenti*, *Phlebotomus papatasi*

## Abstract

This study focuses on identifying the sex and species of sandflies, small insects that thrive in tropical regions and are known to spread diseases such as leishmaniasis and bartonellosis. Traditionally, identifying sandfly species and gender has required a detailed examination of their internal and external anatomy, which is time-consuming and prone to errors. To address these challenges, this research introduces a deep learning model, specifically a convolutional neural network, to automate and improve the accuracy of sandfly identification. Starting with local sandfly samples collected between 2022 and 2023, the study developed a highly efficient model that could significantly reduce the need for manual identification. This approach enhances disease monitoring and control, benefiting public health efforts, especially in regions where sandflies transmit life-threatening diseases. By integrating artificial intelligence into entomology and taxonomy, the study paves the way for faster, more accurate identification methods, with potential implications for improving disease management and prevention worldwide.

## 1. Introduction

A sandfly is a minute insect from the family Psychodidae, commonly found in sandy terrains, especially within tropical and subtropical climates. These insects are notorious for their painful bites and are primarily active during the twilight hours of dawn and dusk, predominantly in rural or wild areas. Unique among psychodids, sandflies have piercing mouthparts that allow them to feed on blood. Due to their small size, typically ranging from 1 to 3 millimeters, they are often hard to spot without magnification. Certain species of sandflies are also vectors for diseases such as leishmaniasis and bartonellosis, which can affect both humans and animals [1]. Identifying specific sandfly species is a complex task that usually requires the detailed examination of their internal anatomy and morphology [2,3]. The sandfly species is generally classified into 23 genera, of which six are the most significant/common: *Phlebotomus*, *Sergentomyia*, and *Chinius* in the Afrotropical, Palaearctic, Malagasy, Oriental, and Australian areas, and *Lutzomyia*, *Brumptomyia*, and *Warileya* in the Nearctic and Neotropical regions [4,5].

Sandflies are well-known vectors for various diseases, which can cause symptoms such as fever, headaches, muscle pain, anemia, skin rashes, and lesions. Typically, it is the female sandflies that are responsible for biting and transmitting diseases, particularly parasitic infections like leishmaniasis. For instance, the World Health Organization reported nearly 90,000 cases of cutaneous leishmaniasis (CL) in war-torn Syria between January and December 2019, with a significant surge in cases in the non-western region [6]. The refugee crisis and the lack of adequate healthcare in these regions highlight the urgent need for further studies on disease vectors like sandflies. Additionally, sandflies are known for their pathogen specificity. For example, *Ph. sergenti* transmits *Leishmania tropica*, while *Ph. alexandri* carries *Leishmania donovani*. Therefore, accurately identifying the species and sex of sandflies is crucial for effective disease control, monitoring, management, and epidemiological research.

The process of determining the gender and species of sandflies typically involves a combination of morphological examination and Internal Anatomy based on well-established identification keys [7,8]. Morphological examination involves differentiating between males and females by the structure of their reproductive organs. Male sandflies have claspers and aedeagi, which are part of their genitalia, while females have a spermatheca, a structure used for storing sperm. Females also possess a pair of piercing mouthparts (proboscis) used for blood-feeding, which are generally more developed than those in males. On the other hand, internal anatomy involves dissecting sandflies to examine internal anatomical features, such as the spermatheca in females or the reproductive organs in males. This process can be integral for distinguishing closely related species because the structure and arrangement of internal organs often provide key diagnostic features.

The aforementioned process can be labor-intensive, time-consuming, and susceptible to misidentification. Additionally, it demands expert knowledge to accurately discern key features from microscopic images. In this work, we introduce and refine a deep learning convolutional neural network (CNN) model for determining the gender and species of sandflies, significantly enhancing the accuracy of taxonomic classification. Integrating such automated methods into existing systems can effectively address the limitations of traditional identification techniques.

The medico-veterinary approach combines the synergy of medical and veterinary sciences to address health challenges at the human–animal–environment interface, under the One Health concept [9]. This multidisciplinary framework is useful for managing zoonotic diseases, such as rabies and leishmaniasis, through collaborative disease surveillance, vector control, and public health interventions [10,11]. Furthermore, advancements in technology and artificial intelligence (AI) enhance the medico-veterinary approach by automating tasks such as disease diagnosis, the species identification of disease vectors, and epidemiological modeling. AI-driven tools, such as convolutional neural networks, have been employed to identify vectors like sandflies and mosquitoes [12,13], which enables accurate and rapid data analysis for disease control [14]. By leveraging these technologies and addressing the impact of environmental changes on disease dynamics, the medico-veterinary approach provides a comprehensive strategy for mitigating global health risks. AI, particularly deep learning, is experiencing significant growth in both the number and variety of applications. The increasing media attention and the broad range of AI-based technologies have further fueled interest in adopting these solutions, leading to greater public acceptance. Fields such as medicine [15], veterinary medicine [16], and smart agriculture [17] are among the many sectors harnessing AI to enhance efficiency, accuracy, and overall process improvement [18]. However, numerous specialized areas remain ripe for development and could greatly benefit from AI technology.

In the field of entomology and taxonomy, deep learning techniques have been increasingly applied to the automatic identification of insect species, with a particular focus on vector mosquitoes due to their importance in disease transmission. One study introduced the InsectNet model, which leverages citizen science data and self-supervised learning for accurate insect species identification with over 96% accuracy across 2500 species, including pollinators, predators, and pests, and has significant implications for pest management and invasive species monitoring [19]. Another study developed an automatic identification system using artificial neural networks (ANNs) and support vector machines (SVMs) for classifying insect specimens at the order level, achieving a classification accuracy of 93% across nine common orders [20]. The ESA-ResNet34 model has also been proposed for crop pest and disease detection, utilizing depthwise separable convolutions to reduce computational load while achieving high accuracy and robustness, with precision, recall, and F1 scores of over 87% [21]. The Yolov8 model for small pest detection in crops demonstrated the highest mean average precision (mAP) of 84.7%, offering real-time pest detection through an integrated Android application [22]. In mosquito vector surveillance, the YOLOv3-based deep learning models have achieved 99% mean average precision and high sensitivity, aiding in accurate gender and species identification for mosquito populations [23]. The VectorBrain CNN model for malaria vector surveillance achieved 94.4% species classification accuracy and 97.7% sex classification accuracy, providing a promising approach for lightweight, mobile-friendly surveillance in field conditions [24]. A separate deep learning model based on AlexNet was employed for yellow fever vector identification, achieving over 90% accuracy, with potential applications in public health surveillance [25]. The use of deep learning for gender identification in crabs was also explored, demonstrating high precision in classifying crabs based on abdominal and shell-side images, with an F1 measure of 98% [26]. Finally, an automated deep learning approach for classifying mosquito species in Korea showed promising results with an F1 score of 97.1%, offering a potential solution for mosquito monitoring and management in the field [27]. There is currently limited research available on the use of AI for the identification of sandflies. In this regard, Cannet et al. [28] employed images of morphological features, such as sandfly wings, to classify sandfly species with an accuracy of 77.0% using a CNN model. In this work, we aim to improve upon their reported accuracy. Moreover, their study did not address the sexing of sandfly species, a gap we intend to explore.

Due to the lack of available public image datasets, this research started from scratch to collect sandfly samples of the local species during the season from May to September of the years 2022 and 2023. These samples were later processed and used to provide us with the input required to develop an efficient and highly accurate model for the sexing and taxonomy of sandfly species. Figure 1 shows a flowchart of the steps used to develop and test the CNN models. In the following sections, we outline each component of the process. Section 2 delves into the dataset, the CNN model, and the performance evaluation metrics and setup. The results are presented and analyzed in Section 3, with the conclusion drawn in Section 4.

## 2. Materials and Methods

### 2.1. Sandfly Data

The sandfly samples were collected from multiple locations in northern and central Jordan over a period of two years. Table 1 shows the number of samples for each classification class considered in this work. A separate data article provides thorough details about the collection process, trap locations, sample preparation protocols, and capture of corresponding images under the microscope [29]. Each sample provided one pharyngeal and one genital image. The individual pharyngeal and genital images were cropped and resized to a manageable 400 × 400 resolution in ’.jpg’ file format, see Figure 2.

### 2.2. CNN Model

Figure 3 shows the baseline CNN model with three 2-D convolution layers. Other variations of the configuration will be discussed in the performance evaluation and comparison in Section 3. The CNN begins with an imageInputLayer(inputSize), which defines the input dimensions of the network. The inputSize parameter specifies the size of the input images, typically represented as Height×Width×Channels. For example, for grayscale images, Channels=1, and for RGB images, Channels=3. For the task at hand, the input image is a fusion of both pharyngeal and genital images. Following this, the network includes a convolutional layer defined as convolution2dLayer(20, 32, ’Stride’, 4). This layer applies 32 filters, each of size 20×20, to the input, with a stride of 4, which reduces the spatial dimensions of the output feature maps, see Figure 4. The output feature map size can be computed using the formula in Equation (Equation 1):(1)$$O=\left(\frac{H-F+2P}{S}\right)+1$$
where *H* is the height/width of the input, *F* is the size of the filter, *P* is the padding (if any), and *S* is the stride.

The batch normalization layer comes after the convolutional layer. This process normalizes the output of the previous activation layer to speed up training and improve stability [30]. The normalized output is given by Equation (Equation 2):(2)$${\hat{x}}^{\left(k\right)}=\frac{x^{\left(k\right)}-\mu^{\left(k\right)}}{\sqrt{\sigma^{2}+\epsilon }}$$
where $x^{\left(k\right)}$ is the activation, $\mu^{\left(k\right)}$ is the mini-batch mean, $\sigma^{2}$ is the mini-batch variance, and ϵ is a small constant added for numerical stability. The output is then scaled and shifted by learnable parameters $\gamma^{\left(k\right)}$ and $\beta^{\left(k\right)}$, respectively, as in Equation (Equation 3):(3)$$y^{\left(k\right)}=\gamma^{\left(k\right)}{\hat{x}}^{\left(k\right)}+\beta^{\left(k\right)}$$

The reluLayer follows, applying the ReLU (Rectified Linear Unit) activation function. ReLU introduces non-linearity into the network by setting all negative values to zero while keeping positive values unchanged. Mathematically, this is represented as in Equation (Equation 4):(4)f(x)=max(0,x)

Finally, a fullyConnectedLayer(numClasses) is added after the convolutional and activation layers, connecting all previous activations to the output neurons. The network concludes with a softmaxLayer, which converts the final layer’s raw output into probability distributions over the target classes. The softmax function for a class *j* is given by Equation (Equation 5):(5)$$\sigma {\left(z\right)}_{j}=\frac{e^{z_{j}}}{\sum_{k=1}^{K}e^{z_{k}}}$$
where $z_{j}$ is the raw output (logit) for class *j*, and *K* is the total number of classes.

Further details of the baseline model layers are shown in Table 2. The model has 429,926 total learnable parameters.

### 2.3. Performance Indicators

The performance of the models was assessed using a variety of metrics designed to highlight their true representative strengths and weaknesses. Equation (Equation 6) illustrates the Matthews correlation coefficient (MCC) when generalized to the multi-class case. The MCC is often regarded as a more insightful and comprehensive performance measure compared to accuracy and F-score [31,32], as it incorporates all components of the confusion matrix into its calculation. Nevertheless, for thoroughness, accuracy (Equation (Equation 7)) and F-score (Equation (Equation 8)) were also included in the results. Additionally, we provided the precision (Equation (Equation 9)), sensitivity (i.e., recall, Equation (Equation 10)), and specificity (Equation (Equation 11)). Each of these metrics exposes different facets of performance. Note that *C* is the confusion matrix and $C_{ij}$ is the confusion matrix element at row *i* and column *j*. Column *i* of the confusion matrix represents the number of times class *i* was predicted (i.e., both true positives and false positives). Conversely, row *i* includes the true number of samples from class *i* (i.e., both true positives and false negatives). The diagonal elements of the confusion matrix represent the number of correct predictions for each class. Each of the metrics, such as precision, recall, and F1-score, represents the average of the corresponding metric calculated for each class.
(6)$$MCC=\frac{\sum_{i=1}^{6}\sum_{j=1}^{6}\sum_{k=1}^{6}\left(C_{ii}C_{jk}-C_{ij}C_{ki}\right)}{\sqrt{\sum_{i=1}^{6}\left(\sum_{j=1}^{6}C_{ij}\right)\left(\sum_{k=1,k\neq i}^{6}\sum_{l=1}^{6}C_{kl}\right)}\sqrt{\sum_{i=1}^{6}\left(\sum_{j=1}^{6}C_{ji}\right)\left(\sum_{k=1,k\neq i}^{6}\sum_{l=1}^{6}C_{lk}\right)}}$$
(7)$$Accuracy=\frac{1}{6}\sum_{i=1}^{6}\frac{TP_{i}}{TP_{i}+TN_{i}+FP_{i}+FN_{i}}$$
(8)$$F-score=\frac{1}{6}\sum_{i=1}^{6}\frac{2TP_{i}}{2TP_{i}++FP_{i}+FN_{i}}$$
(9)$$Precision=\frac{1}{6}\sum_{i=1}^{6}\frac{TP_{i}}{TP_{i}+FP_{i}}$$
(10)$$Recall=\frac{1}{6}\sum_{i=1}^{6}\frac{TP_{i}}{TP_{i}+FN_{i}}$$
(11)$$\mathrm{Specificity}=\frac{1}{6}\sum_{i=1}^{6}\frac{TN_{i}}{TN_{i}+FP_{i}}$$where, *TP_i_* = *C_ii_* is the number of correct predictions of class *i*;${FP}_{i}=\sum_{j=1}^{6}C_{ij}-C_{ii}$ is the number of samples from other classes predicted as class *i*;${FN}_{i}=\sum_{j=1}^{6}C_{ij}-C_{ii}$ is the number of false negative predictions with respect to class *i*;${TN}_{i}=758-{TP}_{i}-{FP}_{i}-{FN}_{i}$ is total number of true negatives with respect to class *i*.


### 2.4. Implementation Details

The base CNN model and its variations were implemented using Matlab R2024a. A unified random seed was used for all evaluation scenarios. To enhance the training process, the images were augmented using the following operations: (1) random reflection over the x-axis; (2) random reflection over the y-axis; (3) random translation along the x and y axes within a pixel range of [−30, 30], allowing shifts left (negative) or right (positive) and up (positive) or down (negative); and (4) random scaling in the x and y directions within the range [0.9, 1.1]. The augmentation process did not increase the dataset size, ensuring no data leakage. Two types of cross-validation were used: 5-fold and 10-fold. In 5-fold cross-validation, the images are randomly divided into five subsets (i.e., folds), each containing 20% of the images. Eighty percent (i.e., four folds) of the dataset is used for training and validation (i.e., seventy percent and ten percent, respectively), and the remaining twenty percent is used for testing as unseen images. On the other hand, 10-fold cross-validation divides the data into 10 equally-sized subsets, with 9 for training and validation and one for testing. In both cross-validation schemes, the training, validation, and testing are repeated from scratch an additional four (5-fold) or nine (10-fold) times so that a different fold is chosen for testing each time. The performance results represent the average and standard deviation from the five (or ten) folds of testing.

The stable and fast converging stochastic gradient descent with momentum (SGDM) algorithm was used to update the network weights during training. Other options are available (e.g., ADAM), but we did not experiment with this parameter. The initial learning rate was set to 0.0003. The minimum batch size was set to 16. The number of training epochs was varied from 50 to 200 in steps of 50. Furthermore, the output of the softmax layer (i.e., class probabilities $p_{j},j=1,\ldots ,6$) is used to calculate the loss function using cross entropy (CE), see Equation (Equation 12), where $t_{i}$ is the target probability for class *i* for a specific image (i.e., an image of class 3 will have $t_{3}=1$ for that image and $t_{j}=0$ for j=1,2,4,5,6).
(12)$$CE\;loss=-\sum_{i=1}^{6}t_{i}\log\left(\frac{e^{p_{i}}}{\sum_{j=1}^{6}e^{p_{j}}}\right)$$

## 3. Results and Discussion

In the CNN architecture, numerous design choices are available. In this study, we focused on the factors that we believed would have a significant and direct impact on performance, including the size and number of filters, the number of layers, and the number of training epochs. Other parameters, such as stride size, downsampling methods, and regularization techniques, were not considered, as incorporating them would have led to an unmanageable number of permutations and detracted from the primary focus of this work.

We begin with the baseline model depicted in Figure 3, which comprises three 2-D convolution layers. The number of filters, consistent across all layers, was selected from the set 4,8,16,32,64. The filter sizes in the convolution layers were [20, 20], [16, 16], and [11, 11] so that features of different granularity are captured. These sizes stem from the sandfly identification keys, which are based on the size and shape of indicative features in genital and pharyngeal images. These features require kernels large enough to capture spatial relationships and morphological patterns but not so large as to include noise or irrelevant information. Conversely, classical convolutional models with small kernels (e.g., 3 or 5) are too fine-grained to effectively capture the distinct morphological structures critical for sandfly gender and species identification. The selected sizes were empirically optimized by evaluating model performance on the task, ensuring an optimal balance between feature extraction and noise reduction. While these experiments were conducted, the detailed results were excluded from the paper to maintain a focused and succinct discussion. This approach allows the network to better align with the unique demands of the dataset and the domain-specific identification criteria. A stride size of 4 was employed. The convolution layers function by sliding a small window across the input image, both vertically and horizontally. As the window moves, it examines the details within each section by multiplying them by a set of values (i.e., the filter) and summing the results, which results in feature maps. No bias term was added. This scanning and calculation process enables the convolutional layer to extract significant features from the image. The filter sizes and stride were carefully chosen based on the size and content of the input images (i.e., sandfly body parts) and the anticipated size of key features in the images, as detailed in the identification keys [7].

Table 3 presents the classification results for all metrics using five-fold cross-validation. Detailed results, including individual performance indices for each fold and the outcomes when using 64 filters, are provided as Appendix A. A significant performance gap is evident when using four filters per 2-D convolution layer, as this configuration is insufficient to capture critical features in the input. Performance appears to peak at 32 filters per 2-D convolution layer. Increasing the number of filters to 64 did not improve performance; instead, training the network with 64 filters per layer for 200 epochs led to overfitting, with a 3% performance drop. At 200 epochs, the results showed MCC@32 = 92.4% compared to MCC@64 = 89.4%, whereas at 150 epochs, they showed MCC@32 = 91.3% compared to MCC@64 = 91.2%. This indicates that the network with 64 filters degraded in performance with additional training, while the one with 32 filters continued to improve from 150 to 200 epochs. For filter sizes smaller than 64, increasing the number of training epochs generally enhances performance, albeit with diminishing returns. These observations are further supported by the confusion matrices in Figure 5, which reveal a higher number of misclassifications (i.e., non-diagonal elements) in the model with 4 filters per layer. In contrast, the confusion matrix for the model with 32 filters per layer demonstrates significantly better classification performance. A similar trend is observed in Table 4, which presents the classification results for all metrics using ten-fold cross-validation, with the full results provided in the Appendix A. The reported performance is slightly higher compared to the five-fold cross-validation, likely due to the increased amount of training data in the ten-fold validation (i.e., 80% of the dataset versus 70%).

Further analysis of the model’s performance was conducted to evaluate the required complexity of the model in terms of the number of layers. To this end, and based on 32 filters per layer, the number of 2-D convolution layers was varied and the network retrained. The different versions were: (1) One 2-D convolution layer with 32 filters of size [20 20]. (2) Two 2-D convolution layers, one with a filter of size [20 20] and the other with a filter of size [16 16], to capture features of various sizes. (3) The baseline design in Figure 3. (4) Four 2-D convolution layers with filters of sizes [20 20], [16 16], [11 11], and [3 3]. (5) Five 2-D convolution layers with filters of sizes [20 20], [16 16], [11 11], [3 3], and [1 1]. Table 5 shows the classification performance versus the number of 2-D convolution layers using five-fold cross-validation. The best performance was achieved with the network consisting of three convolutional layers. Although the four-layer network produced slightly inferior results, it was still competitive, with the two-layer network trailing closely behind. In contrast, the single-layer network was clearly insufficient as it struggled to capture the full range of features in the input, leading to more classification errors. Meanwhile, the five-layer network appeared to overfit the training data, resulting in similarly reduced performance. Figure 6 displays samples of the confusion matrices for the four design options, taken from the final fold of testing.

To gain insight into the decision-making process of the deep learning model, we utilize the gradient-weighted class activation mapping (Grad-CAM) method [33]. This technique highlights, using various colors, the regions of the image that most significantly influence the classification decision. Specifically, the algorithm computes the derivative of the output from the reduction layer for a given class with respect to a convolutional feature map from an appropriate layer (e.g., the final convolutional layer). The regions of the image where this value is high are those that contributed the most to the final classification score. Figure 7 presents two Grad-CAM examples where the synergy between different parts of the image contributed to the correct classification score, compensating for any shortcomings. In the image of the female *Ph. papatasi*, based on the identification keys, the feature map incorrectly emphasized a part of the pharynx that should not have been considered—the tip portion (i.e., the pharyngeal teeth). However, the genitalia map compensated for this error, allowing the algorithm to make the correct decision. Conversely, in the image of the male *Ph. papatasi*, the model correctly identified the relevant part of the genitalia but should have emphasized the entire structure instead of including white space in the top left portion. Additionally, the pharynx area was nearly accurate. The blue regions in the images correspond primarily to white space that does not contain body parts.

Additional analysis was conducted on the role of combining both genital and pharyngeal images in the classification process. Identification keys in the literature emphasize the observation of multiple internal and external morphological features [1,7,8,34]. However, these keys are primarily intended for differentiating among a large number of sandfly species and are designed for human observers, who are prone to errors, limitations, and fatigue and may require multiple clues. To address this, we compared the classification performance when using a single image type (either genital or pharyngeal) versus the fusion of both image types. The comparison was carried out using 10 random runs of 5-fold cross-validation, resulting in 50 testing sets. Table 6 presents the average classification performance when using a single image (pharynx or genitalia) per sample versus the fusion of both images, with five-fold cross-validation repeated ten times, and the baseline model. The superiority of using both images is evident, though genitalia images alone proved highly effective for achieving high classification accuracy compared to pharynx images. Figure 8 displays the MCC results for the 50 testing sets, clearly illustrating the performance differences and superiority. The differences from the results in Table 3 are due to averaging ten times more testing runs in the results from Table 6. The full results are provided in the Appendix A.

There are already many popular CNN models for image classification (e.g., VGG19), which are deployed in many applications in the literature. Table 7 compares four CNN architectures: MobileNet, ResNet101, VGG19, and EfficientNet-b0. Based on performance metrics and computational characteristics, MobileNet stands out with precision of 91.7%, recall of 92.3%, specificity of 98.9%, an F-score of 91.1%, and accuracy of 94.1%, offering a highly efficient solution with only 3.5 million learnable parameters and 154 layers. ResNet101, while deeper with 346 layers and 44.6 million parameters, achieves slightly lower performance metrics, including precision (90.8%), recall (89.3%), and F-score (87.9%), making it less efficient despite its deep architecture that offers capacity for handling complex patterns. EfficientNet-b0 strikes a balance between complexity and performance with precision (89.5%), recall (89.4%), and F-score (87.6%), supported by 5.3 million parameters and 290 layers, but it still falls short compared to MobileNet. However, the best performing custom-built model had 429,926 learnable parameters, which is considered a modest number in comparison to the aforementioned models. This translates into smaller memory/processing requirements and faster execution, which makes it more suitable for real-time applications requiring high performance and low computational cost.

Table 8 shows a summary of recent results on insect identification using machine learning and deep learning methods. The performance comparison table demonstrates the effectiveness of various deep learning models in the classification of species, gender, and other attributes in entomological research. The studies by Lee et al. [27] and Ueki et al. [26] demonstrate strong results with F1 scores of 97.1% and 95%, respectively, for mosquito species classification and horsehair crab gender identification. In comparison, the most relevant study to this work, Cannet et al. [28], report lower performance, with 77% accuracy for phlebotomine sandfly species identification, which reflects the challenges specific to this task. Moreover, the custom-built CNN employed in this work for the identification of the sex and species of phlebotomine sandflies achieved an accuracy range of 95% to 97.1%, positioning it among the better-performing models.

Although the models presented in this work achieved excellent classification performance, several limitations need to be addressed. First, the dataset size (758 images) could be increased and made more balanced to enhance the robustness of the model and produce more consistent results with reduced variability. Second, the Grad-CAM maps revealed that some irrelevant regions, as shown in Figure 7b, were highlighted, which may suggest potential data leakage. However, since the preparation of samples and image acquisition methods were consistent across sample types, further investigation is necessary to understand and eliminate such erroneous decision-making bases. Third, including more sandfly species in the dataset could further improve and generalize the classification model. However, this effort is currently constrained by the locally available species and public datasets. Fourth, the model can be extended to include images of further morphological features (e.g., wing shape). In this case, the extension of image fusion from different body parts needs reevaluation.

## 4. Conclusions

The use of AI is revolutionizing many aspects of daily life, and its applications are becoming increasingly ubiquitous. However, the field of entomology still holds significant potential for harnessing the benefits of AI. In this research, we have successfully investigated the creation of a deep learning AI system capable of accurately identifying the gender and species of the same genus: *Ph. alexandri*, *Ph. papatasi*, and *Ph. sergenti*. By collecting local data and leveraging convolutional neural networks and the combination of genital and pharyngeal images, our system achieved impressive classification performance for several model configurations. This level of precision highlights the significant potential of our method to enhance disease monitoring and control, assist in managing sandfly populations, and support further research and epidemiological studies.

Our findings indicate that deep learning techniques can be effectively utilized in medical entomology, offering an automated and dependable method for sandfly identification. This progress is vital for the research and control of vector-borne diseases, as the precise identification of sandfly species and gender is crucial for comprehending disease dynamics and implementing targeted interventions.

Additionally, our study establishes a solid framework that can be adapted and expanded to other vector species, providing a scalable solution for broader applications in vector-borne disease control and prevention. Future research could investigate the inclusion of additional morphological features and the application of this framework to other medically significant vectors, thereby broadening the impact and usefulness of deep learning in entomological research and public health. Furthermore, the AI model could be integrated into standalone smartphone or microscope applications.

## Figures and Tables

**Figure 1 animals-14-03712-f001:**
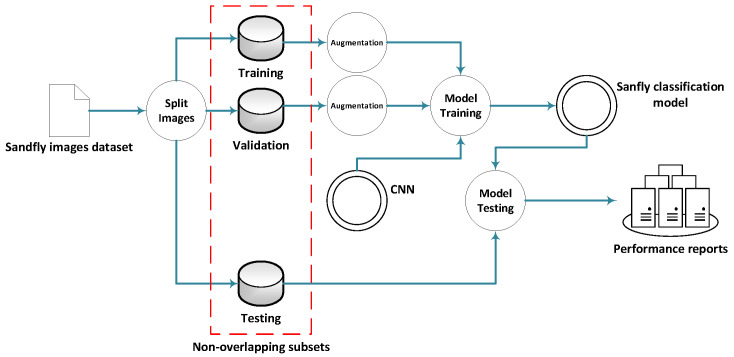
The general steps used to develop the CNN sandfly sexing and taxonomy model.

**Figure 2 animals-14-03712-f002:**
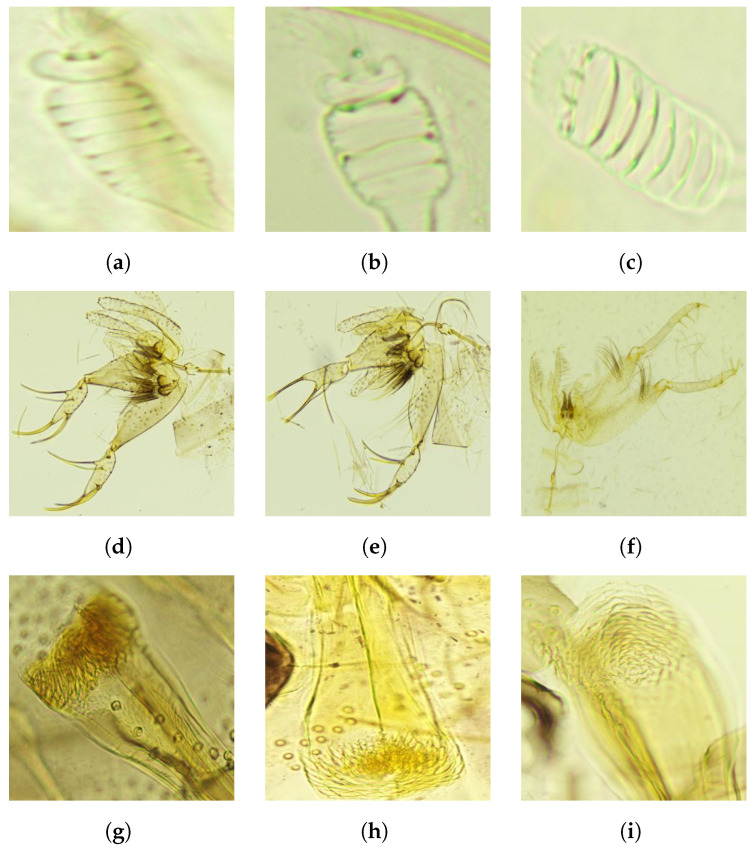
Cropped images from the dataset illustrating the pharyngeal and genital regions, featuring both sexes across the three species. (**a**) *Ph. alexandri* female genitalia. (**b**) *Ph. sergenti* female genitalia. (**c**) *Ph. papatasi* female genitalia. (**d**) *Ph. alexandri* male genitalia. (**e**) *Ph. sergenti* male pharynx. (**f**) *Ph. papatasi* male pharynx. (**g**) *Ph. alexandri* female pharynx. (**h**) *Ph. sergenti* female pharynx. (**i**) *Ph. papatasi* female pharynx. (**j**) *Ph. alexandri* male pharynx. (**k**) *Ph. sergenti* male pharynx. (**l**) *Ph. papatasi* male pharynx.

**Figure 3 animals-14-03712-f003:**
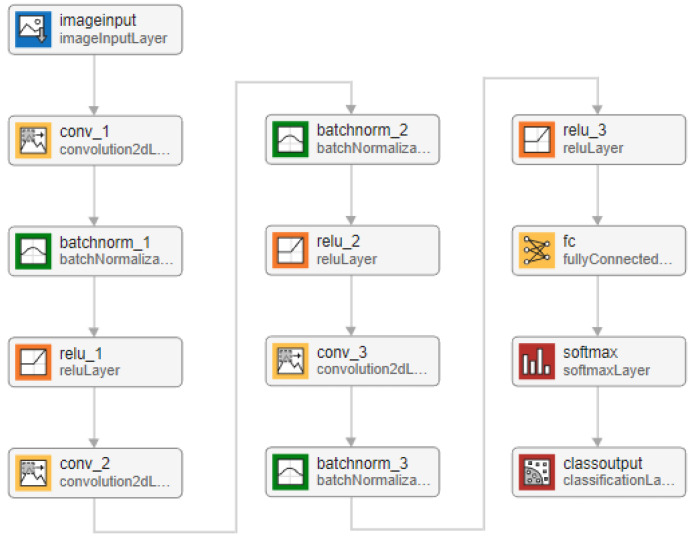
The baseline CNN sandfly identification model. The size of the input to the model is 800×400×3.

**Figure 4 animals-14-03712-f004:**
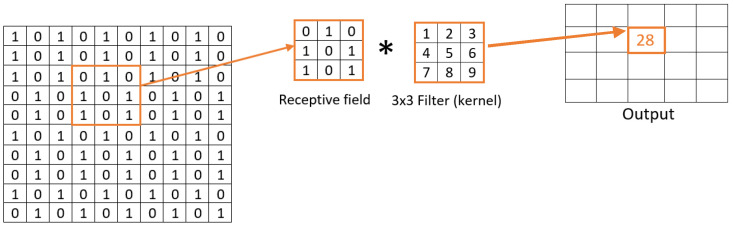
An example of a convolution operation. * denotes convolution operator.

**Figure 5 animals-14-03712-f005:**
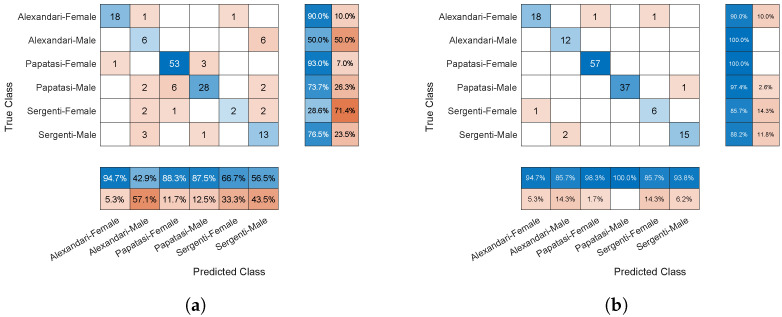
The confusion matrices for the models with 4 filters per layer vs 32 filters per layer, and 200 epochs of training. The matrix is the last result from five-fold cross validation. (**a**) Four filters per layer. (**b**) Thirty two filters per layer.

**Figure 6 animals-14-03712-f006:**
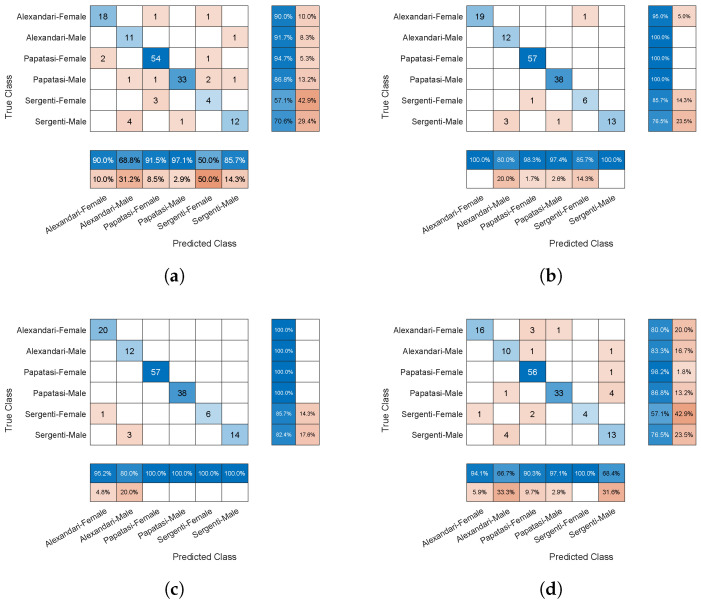
The confusion matrices resulting from the models with differing numbers of 2-D convolution layers and 200 epochs of training. The matrix is the last result from five-fold cross validation. (**a**) One layer. (**b**) Two layers. (**c**) Four layers. (**d**) Five layers.

**Figure 7 animals-14-03712-f007:**
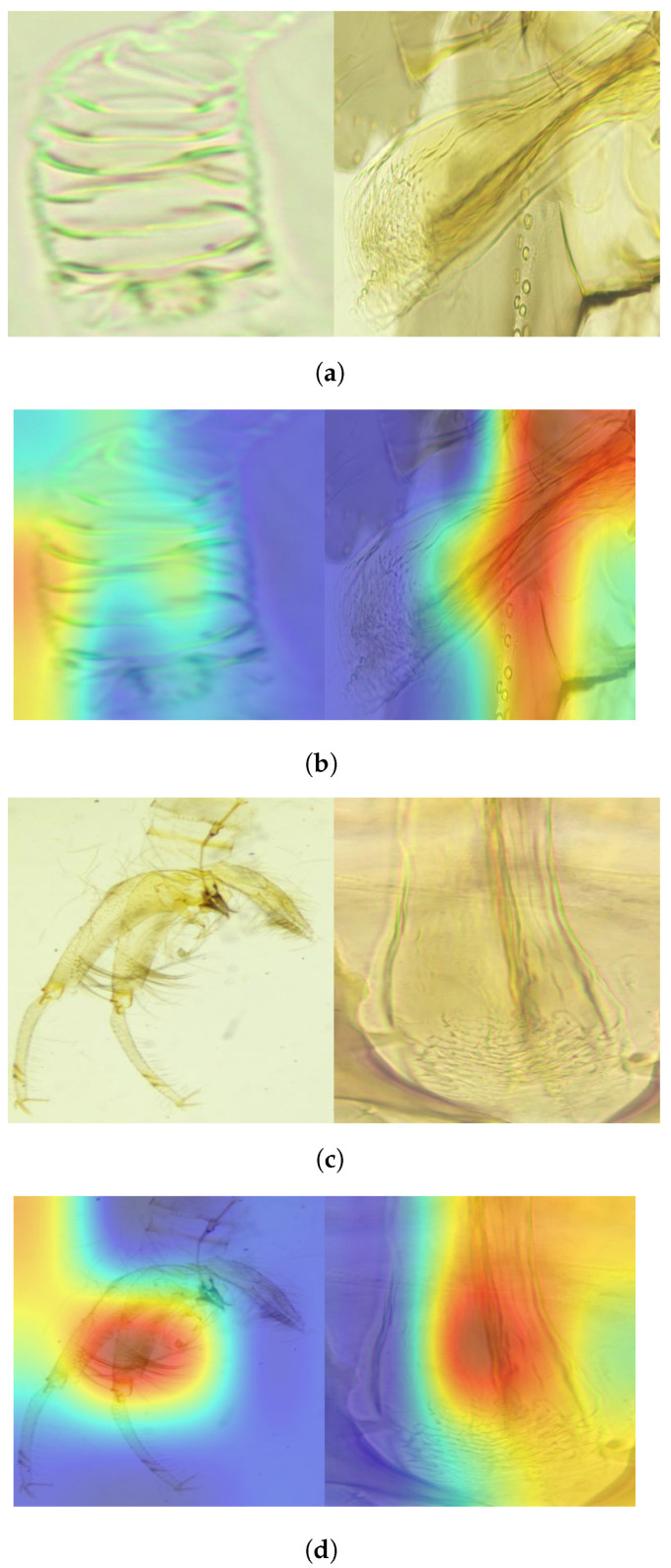
Grad-Cam maps of the respective images showing the importance of different parts of the images in the classification process. The darker red color means that the model pays more attention (i.e., gives more weight) to that part. (**a**) *Ph. papatasi* female Grad-CAM image. (**b**) *Ph. papatasi* female fused image. (**c**) *Ph. papatasi* male fused image. (**d**) *Ph. papatasi* male Grad-CAM image.

**Figure 8 animals-14-03712-f008:**
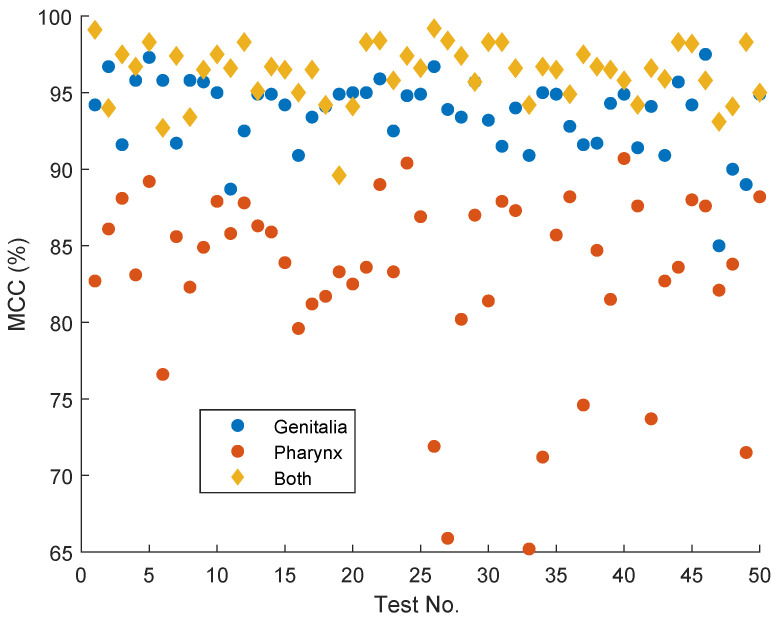
The MCC results from all of the testing folds (i.e., 10 random repetitions of 5-fold cross validation).

**Table 1 animals-14-03712-t001:** Details of the species and sex of the sandfly dataset.

Species	Sex	No. of Samples
*Ph. alexandri*	male	85
*Ph. alexandri*	female	106
*Ph. papatasi*	male	158
*Ph. papatasi*	female	269
*Ph. sergenti*	male	95
*Ph. sergenti*	female	45

**Table 2 animals-14-03712-t002:** Details of the baseline CNN model layers (in order from input). S: Spatial, C: Channel, and B: Batch observations.

Type	Activations	Learnable Sizes	State Sizes
Image Input 400 × 800 × 3 images with ‘zerocenter’ normalization	400(S) × 800(S) × 3(C) × 1(B)	-	-
2-D Convolution 32 20 × 20 convolutions with stride [4 4]	96(S) × 196(S) × 32(C) × 1(B)	Weights 20 × 20 × 3 × 32 Bias 1 × 1 × 32	-
Batch Normalization	96(S) × 196(S) × 32(C) × 1(B)	Offset 1 × 1 × 32 Scale 1 × 1 × 32	TrainedMean 1 × 1 × 32 TrainedVariance 1 × 1 × 32
ReLU	96(S) × 196(S) × 32(C) × 1(B)	-	-
2-D Convolution 32 16 × 16 convolutions with stride [4 4]	21(S) × 46(S) × 32(C) × 1(B)	Weights 16 × 16 × 32 × 32 Bias 1 × 1 × 32	
Batch Normalization	21(S) × 46(S) × 32(C) × 1(B)	Offset 1 × 1 × 32 Scale 1 × 1 × 32	TrainedMean 1 × 1 × 32 TrainedVariance 1 × 1 × 32
ReLU	21(S) × 46(S) × 32(C) × 1(B)	-	-
2-D Convolution 32 11 × 11 convolutions with stride [4 4]	3(S) × 9(S) × 32(C) × 1(B)	Weights 11 × 11 × 32 × 32 Bias 1 × 1 × 32	
Batch Normalization	3(S) × 9(S) × 32(C) × 1(B)	Offset 1 × 1 × 32 Scale 1 × 1 × 32	TrainedMean 1 × 1 × 32 TrainedVariance 1 × 1 × 32
ReLU	3(S) × 9(S) × 32(C) × 1(B)	-	-
Fully Connected	6(C) × 1(B)	Weights 6 × 864 Bias 6 × 1	-
Softmax	6(C) × 1(B)	-	-

**Table 3 animals-14-03712-t003:** The classification results for all metrics using five-fold cross-validation. The numbers represent the average of the five folds of testing.

	100 Training Epochs				150 Training Epochs				200 Training Epochs			
No. of Filters	4	8	16	32	4	8	16	32	4	8	16	32
Precision	68.2%	83.8%	86.7%	91.0%	71.7%	84.1%	88.6%	90.8%	72.2%	88.9%	91.4%	92.8%
Recall	63.8%	75.4%	83.6%	87.1%	65.8%	81.8%	86.8%	89.9%	69.9%	84.8%	91.7%	90.5%
Specificity	93.8%	96.8%	97.9%	98.4%	95.0%	97.7%	98.2%	98.7%	95.8%	98.1%	98.9%	98.8%
F-score	62.8%	77.1%	83.9%	88.1%	66.6%	82.0%	87.2%	89.9%	70.3%	85.5%	91.2%	91.3%
Accuracy	71.6%	85.0%	89.3%	92.2%	75.9%	88.4%	91.2%	93.1%	79.3%	90.6%	94.1%	94.1%
MCC	63.5%	80.7%	86.4%	90.1%	69.0%	85.3%	88.7%	91.3%	73.3%	88.1%	92.4%	92.4%

**Table 4 animals-14-03712-t004:** The classification results for all metrics using ten-fold cross-validation. The numbers represent the average of the ten folds of testing.

	100 Training Epochs				150 Training Epochs				200 Training Epochs			
No. of Filters	4	8	16	32	4	8	16	32	4	8	16	32
Precision	68.7%	82.1%	91.5%	93.9%	73.3%	88.7%	91.4%	93.6%	76.8%	86.7%	92.8%	94.7%
Recall	66.2%	79.0%	89.6%	92.1%	68.7%	84.9%	88.7%	92.6%	74.3%	83.9%	91.8%	94.1%
Specificity	94.2%	97.0%	98.6%	99.0%	95.1%	98.0%	98.6%	99.0%	96.1%	97.9%	98.9%	99.2%
F-score	64.5%	79.1%	90.1%	92.5%	67.8%	85.3%	89.4%	92.6%	73.8%	84.1%	91.7%	94.0%
Accuracy	72.4%	85.5%	93.3%	95.3%	76.8%	90.1%	92.9%	95.0%	81.4%	89.7%	94.3%	96.0%
MCC	64.6%	81.5%	91.4%	93.9%	70.1%	87.5%	90.9%	93.7%	76.3%	86.8%	92.9%	95.0%

**Table 5 animals-14-03712-t005:** The classification performance versus the number of 2-D convolution layers using five-fold cross validation.

Metric\No. of Layers	1	2	3	4	5
Precision	83.9%	91.2%	92.8%	92.4%	83.3%
Recall	84.4%	89.8%	90.5%	90.4%	80.7%
Specificity	97.4%	98.6%	98.8%	98.8%	97.3%
F-score	83.6%	90.0%	91.3%	91.0%	81.0%
Accuracy	87.1%	92.9%	94.1%	94.1%	86.9%
MCC	83.5%	90.9%	92.4%	92.4%	83.2%

**Table 6 animals-14-03712-t006:** The classification performance when using a single image (pharynx or genitalia) per sample vs. the fusion of both images, using five-fold cross validation repeated ten times and the baseline model.

Metric\Input	Genitalia	Pharynx	Fusion of Both
Precision	93.2%	86.5%	95.9%
Recall	92.2%	84.5%	95.8%
Specificity	99.0%	97.2%	99.4%
F-score	92.2%	84.9%	95.7%
Accuracy	95.0%	86.7%	97.1%
MCC	93.7%	83.0%	96.3%

**Table 7 animals-14-03712-t007:** The classification performance when using the fusion of both images and transfer learning.

Model\Metric	Precision	Recall	Specificity	F-Score	Accuracy	MCC	Learnables	Layers
MobileNet	91.7%	92.3%	98.9%	91.1%	94.1%	92.6%	3.5M	154
ResNet101	90.8%	89.3%	98.5%	87.9%	92.1%	90.2%	44.6M	346
VGG19	89.1%	89.0%	98.4%	86.9%	91.2%	89.2%	143.6M	47
EfficientNet-b0	89.5%	89.4%	98.4%	87.6%	91.7%	89.7%	5.3M	290

**Table 8 animals-14-03712-t008:** Summary of recent studies on insect identification using machine learning and deep learning methods.

Study	Problem	Method	Best Performance
Cannet et al. [28]	Species identification of phlebotomine sandflies	MobileNet, ResNet, and YOLOv2	77% accuracy
de Araújo et al. [25]	Identification of yellow fever vectors	AlexNet	94% accuracy
Ueki et al. [26]	Gender identification of horsehair crab	VGG16	95% F1 score
Bjerge et al. [35]	Identify nine insects from trap images	YOLOv5	92.7% precision
Yuan et al. [21]	Crop pest detection	Improved ResNet34	87.1% Precision
Khalid et al. [22]	Small pests detection in field crops	YOLOv8	84.7% mAP
Kittichai et al. [23]	Species and gender identification of mosquito	YOLOv3	99.0% mAP
Pataki et al. [36]	Identification of tiger mosquitoes	ResNet50	96% AUC
Lee et al. [27]	Classification of major mosquito species in Korea	YOLOv5 and others	97.1% F1 score
Li et al. [24]	Identification of mosquito species, sex, and abdomen status	EfficientNet	82.2%–97.7% accuracy
Park et al. [37]	Classification of eight mosquito species	VGG16 and others	97% accuracy
This work	Sex and species identification of phlebotomine sandflies	Custom-built CNN	95%–97.1% accuracy

## Data Availability

The dataset is available from the following link: https://data.mendeley.com/datasets/j9srxj9mvd/2 (accessed on 13 October 2024). The process of capturing the data and details about the dataset are available in the accepted article by Mohammad Fraiwan, Rami Mukbel, and Dania Kanaan. ‘A dataset of sandfly (*Phlebotomus papatasi*, *Phlebotomus alexandri*, and *Phlebotomus sergenti*) genital and pharyngeal images’, data in brief, accepted for publication.

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
