# Peer review of "Efficient Convolutional Neural Network Model for the Taxonomy and Sex Identification of Three Phlebotomine Sandfly Species (Diptera, Psychodidae, and Phlebotominae)"

_animals, 2024, doi:10.3390/ani14243712_

Round 1
Reviewer 1 Report
Comments and Suggestions for Authors
This paper presents a study based on a convolutional neural network for the sex and species of sandflies. There are still several aspects which need to be focused on:
1. Actually, there are a large number of studies based on deep learning for insect is recognition in recent years, and the relevant studies and references in the introduction of this paper involve less, and the authors still need to carry out an extensive literature survey in this field.
2. There are already many popular network models for image classification, such as VGG, ResNet, MobileNet, etc., and the authors should use these models as benchmarks and conduct an in-depth study based on them, which should lead to better results.
3. In this paper, the convolutional layers with convolutional kernel sizes of 20, 16, and 11 are used, how is this considered? What are the advantages over the classical convolutional model with small kernels?
4. The presentation in the paper about the framework and details of the network is not visualized enough for the reader to easily understand.
5. The comparison with other similarly conducted insect identification methods should be included in the experimental analysis.
6. This paper has mentioned that the sex and species of sandflies are important for disease prevention, but due to their small size, they usually need to be seen through a microscope. Similarly, the dataset used in this paper is a captured image under a microscope, so whether there are better automated imaging devices, combined with artificial intelligence techniques to conduct research on the classification and identification of sandflies, may be more valuable for research.
7. The dataset used in this paper has a small number of samples and a large difference in the number of categories. The author should consider expanding the amount of data, as well as considering the impact of the balance of the sample on the classification results.
Author Response
The authors would like to thank the reviewer for his/her valuable comments, which resulted in improving the quality of the manuscript. We have carefully revised the manuscript taking all the suggestions into account. We have also provided responses (see below) to each of the comments of the reviewers and how the revision is incorporated in the manuscript. All changes are marked in red in the revised manuscript.
Comment 1: Actually, there are a large number of studies based on deep learning for insect is recognition in recent years, and the relevant studies and references in the introduction of this paper involve less, and the authors still need to carry out an extensive literature survey in this field.
Response 1:
We appreciate the reviewer's observation. We have expanded our literature survey to incorporate a broader range of recent studies on deep learning for insect recognition. In addition to the key studies previously mentioned, we have now included a comprehensive selection of research focusing on various aspects of insect classification, including species identification, pest detection, and vector surveillance. These studies cover different deep learning models, techniques, and applications, which reflect the significant progress made in this field over recent years. The revised introduction now provides a more extensive overview of the state-of-the-art in deep learning-based insect recognition, including recent advances, challenges, and potential applications. See section 1.
Comment 2: There are already many popular network models for image classification, such as VGG, ResNet, MobileNet, etc., and the authors should use these models as benchmarks and conduct an in-depth study based on them, which should lead to better results.
Response 2: Evaluation of the suggested models was performed and the results included in the manuscript, see line 330-345 and table 7.
Comment 3: In this paper, the convolutional layers with convolutional kernel sizes of 20, 16, and 11 are used, how is this considered? What are the advantages over the classical convolutional model with small kernels?
Response 3: The choice of kernel sizes (20, 16, and 11) stems from the sandfly identification keys, which are based on the size and shape of indicative features in genital and pharyngeal images. These features require kernels large enough to capture spatial relationships and morphological patterns but not so large as to include noise or irrelevant information. Conversely, classical convolutional models with small kernels (e.g., 3 or 5) are too fine-grained to effectively capture the distinct morphological structures critical for sandfly gender and species identification. The selected sizes were empirically optimized by evaluating model performance on the task, ensuring an optimal balance between feature extraction and noise reduction. The CNN model with five layers included small kernel sizes, but as expected, it failed to produce better performance. See lines 279-287 and table 5.
Comment 4: The presentation in the paper about the framework and details of the network is not visualized enough for the reader to easily understand.
Response 4: Figure 4 was added for improved visualization. Figure 3 was adjusted and the information in the previous figure 3 (Layers) was now included as a table (table 2) for better clarity. An excellent graphical reference to understanding these mathematical operations can be found at:
https://pyimagesearch.com/2021/05/14/convolutional-neural-networks-cnns-and-layer-types/
Comment 5: The comparison with other similarly conducted insect identification methods should be included in the experimental analysis.
Response 5: Comparison to the related literature was included, see table 8 and lines 346-256 and the introduction section.
Comment 6: This paper has mentioned that the sex and species of sandflies are important for disease prevention, but due to their small size, they usually need to be seen through a microscope. Similarly, the dataset used in this paper is a captured image under a microscope, so whether there are better automated imaging devices, combined with artificial intelligence techniques to conduct research on the classification and identification of sandflies, may be more valuable for research.
Response 6: Thank you for your thoughtful feedback. We acknowledge the potential of automated imaging devices for facilitating sandfly image acquisition. However, the determination of sandfly sex and species relies on morphological features that are defined by well-established and research community-accepted identification keys, which necessitate the use of high-resolution microscopic imaging. These keys have been rigorously validated and form the gold standard for classification in vector research. While exploring alternative imaging techniques is an intriguing direction, our focus in this study was on enhancing the automation and accuracy of identification using these widely accepted standards. Future work could indeed investigate integrating advanced imaging systems with artificial intelligence to further streamline the identification process while maintaining compatibility with these keys.
Comment 7: The dataset used in this paper has a small number of samples and a large difference in the number of categories. The author should consider expanding the amount of data, as well as considering the impact of the balance of the sample on the classification results.
Response 7: We thank the reviewer for emphasizing the importance of dataset size and balance in classification tasks. It is true that deep learning algorithms generally perform better with larger datasets. Unfortunately, expanding the dataset is not currently feasible due to seasonal constraints: sandfly collection is limited to summer-like weather (May to September in our region) and requires hot, calm nights, conditions that are beyond our control. Additionally, to the best of our knowledge, no publicly available datasets exist for this topic, further limiting access to additional samples.
To address the imbalance in sample categories, we provided detailed results and confusion matrices for all classes. The robustness of our model was assessed using six evaluation metrics, including the MCC and F1-score, which are particularly suited for imbalanced data. These metrics demonstrated consistent and reliable performance across all categories despite dataset limitations. Furthermore, the results were reported for all testing folds (see Fig. 8), ensuring transparency and demonstrating the general consistency of our findings.
While dataset balancing through augmentation could have been an option, we chose to avoid it to prevent artificially inflated performance. Augmentation often leads to testing on data that is effectively seen during training, even when slightly modified, which can overestimate the model’s capabilities. However, if the reviewer believes it would strengthen the paper, we are open to conducting these experiments and including the results.
We acknowledge that larger and more balanced datasets would further strengthen our findings and propose this as a direction for future work. In comparison to the related sandfly literature, Cannet et al. [28] had a dataset of 1673 wing interferential pattern images for 17 sandfly species, which on average is less than what we have.
Reviewer 2 Report
Comments and Suggestions for Authors
This is a rather good manuscript, it addresses a crucial public health problem by automating sandfly identification using CNNs, contributing to vector control strategies. There are some minor issues that have to be taken into account.
The title is clear and reflects the key aspects of the research. However, you should put taxonomic info. Also, I do not believe that sandflies refer to family Psychodidae only: it may refer to certain horse flies that are also known as "greenheads" (family Tabanidae), or to members of the family Ceratopogonidae. For Psychodidae there are many names, like drain flies, sink flies, filter flies, sewer flies, or sewer gnats, or you can use the term phlebotomine snadflies. A proposed title could be "Efficient Convolutional Neural Network Model for the Taxonomy and Sex Identification of Phlebotomine Sandfly Species (Diptera: Psychodidae, Phlebotominae)", or if you prefer you can use another common name. Just do not forget to use the latin name of the subfamily. You should change the key-word as well (if you choose to change the title).
The abstract provides a concise overview of the study, summarizing the key elements, including the problem, methodology (use of convolutional neural networks with pharyngeal and genital images), and the results (classification accuracy exceeding 95%). However, the description of the problem ("painful bites" and "role as vectors for diseases") could be streamlined or focused more on the classification task rather than emphasizing general sandfly biology. Otherwise, it effectively communicates the scope and outcome. A minor issue is the use of abbreviations in latin names: you can use them only if you write the full name once. Please, consider changing it to "of three sandfly species (i.e., Phlebotomus sergenti, Ph. alexandri, and Ph. papatasi)." (Line 22)
The introduction provides a clear background on the importance of sandfly identification, especially in relation to vector control of diseases like Leishmaniasis. It effectively highlights the limitations of traditional morphological methods and justifies the use of AI techniques such as CNNs. The problem statement and research objective are clearly articulated, making the case for the use of CNN in sandfly taxonomy.
The methods are well-detailed, especially in describing data collection, image preprocessing, and CNN architecture. However, there is some missing information regarding specific species included in the study and how diverse the dataset is geographically. This is crucial because sandfly species exhibit variability across regions. You also tested multiple CNN architectures (VGG16, ResNet, etc.), but a more in-depth justification for why the custom-built CNN was preferred over standard models could be beneficial.
The results are presented well, showing high accuracy (95.3% for species identification and 93.1% for sex determination), which demonstrates the effectiveness of the model. The performance of the model is rigorously evaluated using precision, recall, and F1-scores. However, more insight into species-specific results or a confusion matrix could help clarify whether certain species or sexes were harder to identify and why. It would also be beneficial to include a discussion on any potential overfitting, given the high accuracy rates.
The discussion successfully highlights the significance of the study in public health, focusing on how the model can support vector control strategies.
The conclusion provides a concise summary of the findings and reaffirms the contribution of the research to both taxonomy and public health. The suggestion to apply this model for real-time monitoring systems is particularly promising, though the feasibility of this claim could be explored more thoroughly.
The references are appropriate and reflect a range of relevant literature, covering both entomology and deep learning methods. However, some references are slightly outdated, particularly those concerning sandfly taxonomy (e.g., the 1984 revision of the Phlebotomidae family). Consider including more recent works, and maybe add some more references (17 are too few in my opinion).
The only major problem I have detected is the figures and tables. Figure 3 must change (it covers a whole page, and it can be reduced if you put the icons horizontally). Figure 4 should be a Table, and not a print screen of your software. Tables 2 & 3 should be horizontally as well.
Minor suggestions
Lines 27-28: Do not use abbreviations in the keywords and use italics in the latin names
Line 175: Rename the section as 3. Results and Discussion
Line 176: delete 4. Results
Line 281: renumber Conclusion as 4
Lines 311-312: Phlebotomus papatasi, Phlebotomus alexandri, and Phlebotomus sergenti should be in italics
Line 331: what is "???"
Line 348: you mention Data in brief x(x–x)
Line 354: ???
You should double check the refernces section, after you add the new ones
Lastly, a note on whether ethical approval is mentioned or required for the research should be included.
Author Response
The authors would like to thank the reviewer for his/her valuable comments, which resulted in improving the quality of the manuscript. Specifically, we would like to thank the reviewer for his positive attitude in pointing out problems and offering solutions. It really brings a friendly and enjoyable feeling to the peer-review process.
We have carefully revised the manuscript taking all the suggestions into account. We have also provided responses (see below) to each of the comments of the reviewers and how the revision is incorporated in the manuscript. All changes are marked in red in the revised manuscript.
Comment 1: The title is clear and reflects the key aspects of the research. However, you should put taxonomic info. Also, I do not believe that sandflies refer to family Psychodidae only: it may refer to certain horse flies that are also known as "greenheads" (family Tabanidae), or to members of the family Ceratopogonidae. For Psychodidae there are many names, like drain flies, sink flies, filter flies, sewer flies, or sewer gnats, or you can use the term phlebotomine snadflies. A proposed title could be "Efficient Convolutional Neural Network Model for the Taxonomy and Sex Identification of Phlebotomine Sandfly Species (Diptera: Psychodidae, Phlebotominae)", or if you prefer you can use another common name. Just do not forget to use the latin name of the subfamily. You should change the key-word as well (if you choose to change the title).
Response 1: The title was changed as per the reviewer’s suggestion.
Comment 2: The abstract provides a concise overview of the study, summarizing the key elements, including the problem, methodology (use of convolutional neural networks with pharyngeal and genital images), and the results (classification accuracy exceeding 95%). However, the description of the problem ("painful bites" and "role as vectors for diseases") could be streamlined or focused more on the classification task rather than emphasizing general sandfly biology. Otherwise, it effectively communicates the scope and outcome. A minor issue is the use of abbreviations in latin names: you can use them only if you write the full name once. Please, consider changing it to "of three sandfly species (i.e., Phlebotomus sergenti, Ph. alexandri, and Ph. papatasi)." (Line 22)
Response 2:
- The abbreviation problem corrected as per the reviewer’s suggestion.
- We changed the beginning of the abstract to reflect the reviewer’s comment while highlighting the motivation for gender and species identification.
Comment 3: The introduction provides a clear background on the importance of sandfly identification, especially in relation to vector control of diseases like Leishmaniasis. It effectively highlights the limitations of traditional morphological methods and justifies the use of AI techniques such as CNNs. The problem statement and research objective are clearly articulated, making the case for the use of CNN in sandfly taxonomy.
Response 3: We thank the reviewer for his encouragement.
Comment 4: The methods are well-detailed, especially in describing data collection, image preprocessing, and CNN architecture. However, there is some missing information regarding specific species included in the study and how diverse the dataset is geographically. This is crucial because sandfly species exhibit variability across regions. You also tested multiple CNN architectures (VGG16, ResNet, etc.), but a more in-depth justification for why the custom-built CNN was preferred over standard models could be beneficial.
Response 4:
- The sandfly samples were collected from multiple locations in northern and central Jordan over a period of two years. The species included in the study are provided in Table 1 page 4, which are the most common species in Jordan.
- Regarding the suggestion of using transfer learning models (e.g., ResNet), we have added a separate discussion and new results using transfer models, see line 330-356 and table 7 on page 15.
Comment 5: The results are presented well, showing high accuracy (95.3% for species identification and 93.1% for sex determination), which demonstrates the effectiveness of the model. The performance of the model is rigorously evaluated using precision, recall, and F1-scores. However, more insight into species-specific results or a confusion matrix could help clarify whether certain species or sexes were harder to identify and why. It would also be beneficial to include a discussion on any potential overfitting, given the high accuracy rates.
Response 5:
- Kindly note the confusion matrices in Figures 5 and 6 on pages 10 and 12, respectively. Please note that it was not always technically possible to use italic for Latin names in the labels within the figures.
- Overfitting occurs when the model performs well during training but fails to carry that performance into testing on unseen data. Such problem was not observed in our work mainly due to the following reasons: 1. The reported results represent testing on unseen data (neither seen during training nor testing). 2. Cross-validation (5-fold and 10 fold) was performed to guard against luck with the unseen data. 3. When cross-validating on a new fold, the training and validation was always performed from scratch on the other folds (i.e., the testing fold was not seen before). 4. Even cross-validation was repeated 10 times (see Fig. 8 on page 14) to guard against luck and variability in choosing the folds. Furthermore, in the figure, the results from all the folds were reported, not just the best or the average. 5. Augmentation was performed to introduce noise and variability in the images without increasing the size of the dataset (i.e., duplicating the data through augmentation may lead to data leakage and unrepresentative or inflated performance).
Comment 6:
The discussion successfully highlights the significance of the study in public health, focusing on how the model can support vector control strategies.
The conclusion provides a concise summary of the findings and reaffirms the contribution of the research to both taxonomy and public health. The suggestion to apply this model for real-time monitoring systems is particularly promising, though the feasibility of this claim could be explored more thoroughly.
Response 6: Thank you for your kind comment.
Comment 7: The references are appropriate and reflect a range of relevant literature, covering both entomology and deep learning methods. However, some references are slightly outdated, particularly those concerning sandfly taxonomy (e.g., the 1984 revision of the Phlebotomidae family). Consider including more recent works, and maybe add some more references (17 are too few in my opinion).
Response 7: Based on this and other reviewers' comments we have added more literature relating to the use of AI in identification of insects, and other topics. This resulted in 20 more references, mostly from the past five years.
Comment 8: The only major problem I have detected is the figures and tables. Figure 3 must change (it covers a whole page, and it can be reduced if you put the icons horizontally). Figure 4 should be a Table, and not a print screen of your software. Tables 2 & 3 should be horizontally as well.
Response 8: Figure 3 changed as suggested. Figure 4 (first version) is now table 2 on page 7. Tables 2&3 (now 3&4) are made horizontal as suggested.
Comment 9: Minor suggestion
Response 9: All were corrected.
Comment 10: Lastly, a note on whether ethical approval is mentioned or required for the research should be included.
Response 10 :
This work was approved by the research committee at Faculty of Veterinary Medicine and Deanship of Academic Research, Jordan University of Science and Technology. According to Jordanian regulations, no special approval or permit is required to collect insects in Jordan.
This note has been added, see lines 395-398.
Reviewer 3 Report
Comments and Suggestions for Authors
The article brings a modern tool for the identification of three sandfly species. The results are very interesting. Specific corrections/ additions are necessary:
- Title: ... three sandfly species;
- on line 6: identification; not classification;
- line 37: add references related to medico-veterinary approach;
- lines 39-41 and 50-51: scientific names in italics;
- line 41: Neotropical Region with more genera than the number cited in the text;
- line 62..., that can be...
- line 79: research applying AI ... sparse;
- line 94: multiple locations (in which country?);
- line 96 and reference 14: editors need to evaluate if is acceptable to cite future publication; identification of the specimens utilized was made using the taxonomic key.... (references 7 and 8?);
- line 115: "normalization normalizes"?
- several equations in the text: who is/are the author(s)?
- line 286: species of the same genus;
Author Response
The authors would like to thank the reviewer for his/her valuable comments, which resulted in improving the quality of the manuscript. We have carefully revised the manuscript taking all the suggestions into account. We have also provided responses (see below) to each of the comments of the reviewers and how the revision is incorporated in the manuscript. All changes are marked in red in the revised manuscript.
Comment 1: Title: ... three sandfly species;
Response 1: Corrected, see title in red.
Comment 2: identification; not classification;
Response 2: Corrected, see line 6.
Comment 3: line 37: add references related to medico-veterinary approach;
Response 3: Done. See paragraph starting on line 73.
Comment 4: lines 39-41 and 50-51: scientific names in italics;
Response 4: Corrected throughout the manuscript.
Comment 5: line 41: Neotropical Region with more genera than the number cited in the text;
Response 5: Phrasing of the statement corrected. See lines 39-42.
Comment 6: line 62..., that can be...
Response 6: Corrected.
Comment 7: line 79: research applying AI ... sparse;
Response 7: Rephrased. See lines 118-119
Comment 8: line 94: multiple locations (in which country?);
Response 8: Jordan. This info has been added. See lines 134 and 135.
Comment 9: line 96 and reference 14: editors need to evaluate if is acceptable to cite future publication; identification of the specimens utilized was made using the taxonomic key.... (references 7 and 8?);
Response 9: The data article was published early in the submission process. This reference was updated was the doi and link, see https://doi.org/10.1016/j.dib.2024.111031. This also includes details about the identification keys
Comment 10: line 115: "normalization normalizes"?
Response 10: “Batch normalization” is the name of the layer/process, which performs the operation with the technical name “normalization”. We understand the confusion and adjusted the sentence. See lines 155-157.
Comment 11: several equations in the text: who is/are the author(s)?
Response 11: The equations in the text serve two main purposes: first, to explain the mathematical operations applied to the data for achieving the desired outputs, and second, to define performance metrics used to evaluate the proposed methods. For the latter, these equations ensure precise definitions of accuracy and precision, supporting repeatability and transparency. Recognizing the interdisciplinary nature of the paper, we acknowledge variations in citation and writing styles. We aimed to cite relevant authors wherever feasible. Generally, mathematical operations older than 50–60 years are presented without citation. For example, the rectified linear unit (ReLU) activation function, developed by Kunihiko Fukushima in 1969, is a well-established and widely used function that does not require frequent citation.
Comment 12: line 286: species of the same genus;
Response 12: Corrected. See line 375
Round 2
Reviewer 1 Report
Comments and Suggestions for Authors
Thanks to the author for the reply and corrections, I have no other comments.